# Conscious Mobility for Urban Spaces: Case Studies Review and Indicator Framework Design

Roberto C. Vargas-Maldonado [1], Jorge G. Lozoya-Reyes [1], Mauricio A. Ramírez-Moreno [1,†], Jorge de J. Lozoya-Santos [1,*], Ricardo A. Ramírez-Mendoza [1], Blas L. Pérez-Henríquez [2], Augusto Velasquez-Mendez [3], Jose Fernando Jimenez Vargas [3] and Jorge Narezo-Balzaretti [4]

1 School of Engineering and Science, Tecnologico de Monterrey, Eugenio Garza Sada 2501 Sur, Monterrey 64849, Mexico
2 Mexico Clean Economy 2050, Precourt Institute for Energy, Stanford University, Stanford, CA 94305, USA
3 Department of Electrical and Electronic Engineering, Universidad de los Andes, Bogota 111711, Colombia
4 MSc Regional and Urban Planning Studies, The London School of Economics and Political Science, Houghton St., London WC2A 2AE, UK
* Correspondence: jorge.lozoya@tec.mx

**Abstract:** A lack of data collection on conscious mobility behaviors has been identified in current sustainable and smart mobility planning, development and implementation strategies. This leads to technocentric solutions that do not place people and their behavior at the center of new mobility solutions in urban centers around the globe. This paper introduces the concept of conscious mobility to link techno-economic analyses with user awareness on the impact of their travel decisions on other people, local urban infrastructure and the environment through systematic big data collection. A preliminary conscious mobility indicator framework is presented to leverage behavioral considerations to enhance urban-community mobility systems. Key factors for conscious mobility analysis have been derived from five case studies. The sample offers regional diversity (i.e., local, regional and the global urban contexts), as well as different goals in the transformation of conventional urban transport systems, from improving public transport efficiency and equipment electrification to mitigate pollution and climate risks, to focusing on equity, access and people safety. The case studies selected provide useful metrics on the adoption of cleaner, smarter, safer and more autonomous mobility technologies, along with novel people-centric program designs to build an initial set of conscious mobility indicators frameworks. The parameters were applied to the city of Monterrey, Nuevo Leon in Mexico focusing on the needs of the communities that work, study and live around the local urban campus of the Tecnologico de Monterrey's Distrito Tec. This case study, served as an example of how conscious mobility indicators could be applied and customized to a community and region of interest. This paper introduces the first application of the conscious mobility framework for urban communities' mobility system analysis. This more holistic assessment approach includes dimensions such as society and culture, infrastructure and urban spaces, technology, government, normativity, economy and politics, and the environment. The expectation is that the conscious mobility framework of analysis will become a useful tool for smarter and sustainable urban and mobility problem solving and decision making to enhance the quality of life all living in urban communities.

**Keywords:** transportation systems; urban mobility; sustainable mobility; smart mobility; electric mobility; mobility behaviors; conscious mobility, conscious technologies; conscious community

## 1. Introduction

Transportation systems are important elements to take into account when trying to improve the quality of life in cities, as they structure many of the social and economic interactions that take place every day in urban areas. When transport systems are efficient, they provide economic and social opportunities and benefits that result in positive multiplier

effects such as better accessibility to markets, employment, and additional investments [1]. However, as cities become more attractive, new residents arrive, compromising the design capacity of road infrastructure and public transportation. Moreover, urban population growth leads to new housing development, which, in turn, could result in urban sprawl.

The pace of urban expansion is expected to increase in the next decades, and this represents a challenge for the delivery of urban passenger transport services facing a projected growth of 60–70% by 2050 [2,3], at the same time, motorized transport is expected to increase 94% between 2015 and 2050 [2]. This path leads to a 26% global increase in $CO_2$ emissions by urban activity alone [2]. These problematic projections are driven by a combination of insufficient sustainability considerations in urban mobility plans, and not taking enough advantage of novel technologies (e.g., digitization, data mining, artificial intelligence and optimization, etc.) that hampers the level of understanding of urban communities transportation needs and quality of life aspirations. The identification of these issues has encouraged the public and private sectors to pursue new mobility strategies, such as sustainable and smart mobility, where the objective is to make all transport modes more sustainable, make sustainable alternatives widely available in a multi-modal transport system, and put in place the right incentives to drive this transition [4].

According to [5–7], smart mobility, plans to implement actions and reduce pollution and waste while increasing transport efficiency [8]. Therefore, smart mobility is deeply connected to the performance of sustainable mobility, contributing to improving the quality of life [9]. Smart mobility is an integrated system comprised of several projects and actions, all aimed at sustainability in urban development [10]. Therefore, even though the smart mobility concept is evolving, sustainability should be considered as inherent to it, thus any new transport modes that seek to be called "smart", must be resource efficient, balancing environmental needs with those of the economy and the community. In general, smarter communities are those that adopt novel digital and clean technologies for urban and transport infrastructure systems problem solving through business, policy and social innovations that aim to enhance quality of life for all [11].

New mobility modes are desired to enhance and better integrate intermodal urban transport systems. In the past decade, smart mobility has started to emerge as an innovative approach vis-à-vis traditional transport modes, example assisted/self-driving, intelligent, logistics, smart parking, and sharing economy (ride-/car-/bike-sharing) [12]. Besides development in new power source technologies, such as compressed natural gas (CNG), hybrid electric, and full electric, represent the near future of road passenger transport. Replacing diesel buses with CNG buses in urban transport was part of the first attempts from urban planners and decision makers, as it reduces the emission of toxic substances and greenhouse gases and contributes to reducing the negative impact from the transportation sector on the environment [13]. However, based on the UK data, results showed that HydrogenFuel Cell Vehicles (FCVs) will come to dominate, but only in the very long run (after 2030), while biofuels and ICE(Internal Combustion Engine)-electric hybrids will be the main alternatives to the regime in the next 10–30 years, because (a) they are already developed and (b) they fit better into current infrastructures [14].

Governments around the world are starting to pass legislation ban fossil-fueled combustion engines to advance their climate commitment under the United Nations´ Paris Agreement to address global climate change. Thus, an increasing number of organizations, companies, policymakers, and the general public are focusing their attention on the subject of electric mobility [15]. Electric vehicles (EVs) have become a relevant topic of discussion, promoting electric cars and electric buses (EBs) as the key products to advance a clean economy and achieve the goal of net zero emissions from the transport sector. Thus, as effective congestion management requires extensive use of public transport, the introduction of zero-emission vehicles for private and public, can help reduce pollution emissions immediately. Nevertheless, in terms of policies and their implementation, transition is complex, with a series of emerging societal implications, that government need to understand, and address [16].

Lack of understanding of the positive outcomes of these technologies is due in part to the lack of more holistic methods of analysis and not taking advantage of the of big data, particularly of digital sources of consumer behavior and urban mobility. In addition, many decision-makers still rely heavily on technocentric solution. Moreover, many transport users are not aware of the impact that their travel decisions have on the transportation system. Increasing travelers' awareness of the environmental impact of travel mode choices and changing the citizen's behavior toward adopting transportation habits that rely more on the use of public transportation, bicycles, and walking, and less on private cars can provide the means to reduce GHG emissions and co-pollutants, thus mitigating the effects on the environment and the planet [17].

To increase this environmental awareness and assess the level of consciousness behind urban mobility in cities, we suggest the concept of conscious mobility. The main goal of this conceptualization is to create a shift in citizens' travel behavior by implementing technologies that allow the users to understand which are the sustainable travel decisions that fit with their necessities and also demand and foster the provision of transport modes that meet these criteria towards smarter, more sustainable communities. However, to be implemented, tools for the evaluation and establishment of relationships between conscious mobility projects should be created, in order to facilitate its acceptance among urban planners and decision-makers in the mobility space. Community engagement and public consultation is critical to the pertinence and political acceptance of any infrastructure transformation project, particularly as we transition to more sustainable urban mobility systems.

Ideally, an integrated approach balancing multiple criteria is beneficial for comprehensive city evaluation and wide usability and acceptance of an indicator standard [18]. The European Telecommunication Standards Institute (ETSI) and International Telecommunication Union (ITU) are in charge of unifying technology and telecommunications standards in topics such as mobility and transportation of inhabitants and goods. Thus frameworks ETSI 103463 and ITU 4903 provide the best balance with the criteria considered in this study. In this way, the conscious mobility framework was developed, unifying the sustainable-smart mobility strategies through smart cities frameworks [19,20], regarding dimensions such as society and culture, infrastructure and urban spaces, technology, normativity, economy and politics, and the environment.

Each project of conscious mobility follows the same performance dimensions (i.e., cleaner and more sustainable), however, it can pursue different objectives, hence analyzing initiatives from different regions and communities led to a more holistic perspective of what metrics should be assessed as part of the proposed framework. Given the lack of research and projects about travel decision behavior and sustainable transport modes, especially in Latin America, the cities of Bogota, Ohio, and Mexico were assessed, gathering valuable insight about which challenges and opportunities should be considered for the success of conscious mobility projects. Shenzhen was selected as the global benchmark due to its massive electrification and clean economy success. The main case study as described below was selected for its deep contextual understanding by this research group of the local needs, facilitating the pilot application of the conscious mobility framework for a specific community and urban region.

Distrito Tec, is an urban campus in the city of Monterrey, State of Nuevo Leon, Mexico was selected as the main case study, specifically because of the Distrito Tec urban regeneration initiaitve driven and fostered by the Tecnologico de Monterrey (ITESM) university campus, in collaboration with residents, authorities, and organizations [21] (A better overview of the geographical context is presented in Figure 1). A preliminary analysis of the conscious mobility dimensions, status, and stakeholder involvement was conducted, using key projects efforts and current strategies deployed to date, to better understanding the goals pursued and their actual implementation.

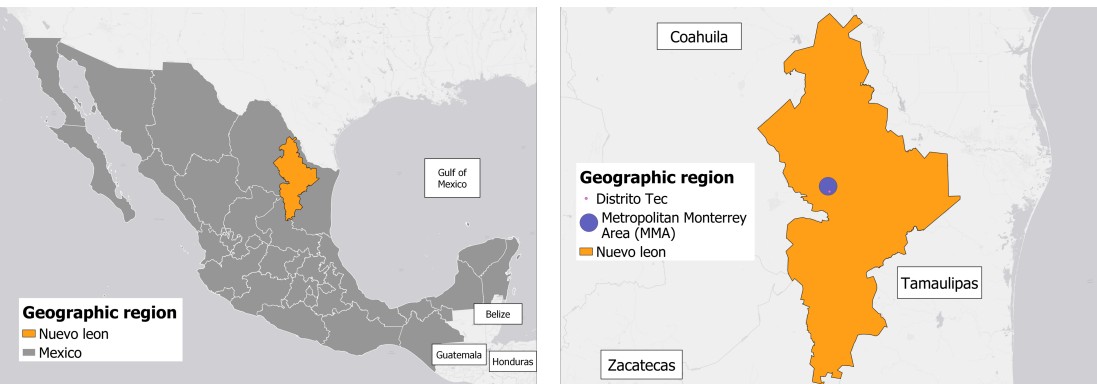

**Figure 1.** Case study geographical location.

The main objective of this research is to define the concept of conscious mobility using a predetermined set of indicators in a selection of contexts of analysis. Using such research design strategy, this paper reviews and assess under the conscious mobility framework a sample of the representative transport projects of electrification. The central research question is to what extent is the city of Monterrey prepared to embrace conscious mobility? Ultimately, the aim of this study is to present a more holistic framework of analysis for cleaner, more sustainable and smarter mobility plans in other Mexican or Latin American cities and districts. Given the cultural, technological, and political characteristics of each city, the analysis and indicators used should not be regarded as a fixed or standardized, as the key variables to assess correspond to the goals set by the authors of the research to the specific context of DistritoTec urban campus. But serve as the start of a research agenda.

## 2. Conscious Mobility

Sustainability is composed of three main factors: society, economy, and environment [22]. The first factor, "society", focuses on perceiving the satisfaction of the needs of the population, its fulfillment, and the distribution of that fulfillment, social equity. On the other hand, economics is related to opportunities and benefits that result from access to or increased accessibility of markets, employment, and sources of credit finance for large livelihood costs, potentially including higher education, housing, transportation, and entrepreneurship. Finally, the environment concerns the footprint of human activities and their relationship with environmental systems and the intergenerational responsibility of common future report [23] The concept of sustainable mobility takes into account these factors by applying them in a mobility system. Decision makers are also cognizant of emerging technologies such as, compressed natural gas buses, transport electrification, digitization automation enabled or incentivized by public policy to ensure widespread adoption within society.

The rapid growth of Information and Communication Technologies (ICT) is a major factor in actualizing smart mobility for many countries because they can use these now widely available technologies to better manage transport systems [24]. Digitization, interoperability, and open information are needed to help these cycles of open metropolitan advancement in an ICT-empowered city [24]. Technocentric approaches to smart cities are the most commonly used approaches in the literature, but they can result in solutions that are prejudiced against sustainability goals [25,26]. Therefore, in addition to analyzing solutions from a technological perspective, the approach to smart cities should analyze sustainability factors that require attention to the particular contexts of each city [27].

To fully develop context-aware, locally sustainable, and equitable mobility system, we suggest adding a consciousness factor to traditional sustainable mobility approaches (see Table 1). Firstly, this approach requires mobility strategies to cultivate citizens and foster positive behavior of residents who recognize and comprehend the impact of their decisions in urban environments, allowing them to fulfill their daily needs while having as little

impact as possible on society and the environment (i.e, minimize negative externalities). This is what we call a conscious community. Secondly, this approach requires technology that enables, supports and improves the relationship between citizen behaviors and the urban environment, identifying opportunity areas to increase benefits by upgrading urban structures based on user needs and sustainable development goals (SDGs). This is what we denote as conscious technologies. The integration of these two aspects with the current sustainable mobility is what we recognize as ***conscious mobility***.

**Table 1.** Mobility strategies comparison.

| Type of Mobility | Definition | Approach | Dimensions |
|---|---|---|---|
| Sustainable mobility | Include a better provision of infrastructure and services to support the movement of goods and people [28] | SDGs as the basis of urban mobility | Society, Economy and Environment |
| Smart mobility | Smart mobility contains a number of actions that enhance users' mobility by foot, public or private transportation, or any other means of transport. It leads to a reduction in economic costs that are incurred by the environment and time [24] | Incorporation of technology and use of SDGs to solve transportation issues | Society, Economy, Environment, and Technology |
| Conscious mobility | Mobility leveraged by conscious communities capable of incorporating conscious technologies for the development of sustainable-smart transportation systems | Citizen travel decision, technology embracement, and conscious infrastructure design | Culture and Society, Infrastructure and urban spaces, Technology, Policy, Normativity and Economy, and Environment |

We defined conscious mobility as a type of mobility leveraged by conscious communities capable of incorporating conscious technologies for the development of sustainable-smarter transportation systems. There are several modes of conscious mobility that we identify, including cycling and motorized transport by cleaner fuel sources, but the integration of conscious technologies is what drives scale in adoption. Conscious technologies can be classified into four types, such as data gathering, simulation, analysis and evaluation, and application; the last one can be divided into primary and supportive technologies. This set of technologies is meant to be used together or separately, helping to embrace a successful behavioral change in citizens towards clean, smarter transportation alternatives.

In this context, conscious technologies can also be called accessible and persuasive technologies, tailored for and integrated into applications that support mobility (e.g., route planners), which can affect travelers' decisions and available choices, and guides them toward selecting routes that are environmentally friendly. Persuasive technology is broadly defined as technology that is designed to change the attitudes or behaviors of the users through persuasion and social influence but not through coercion [17]. The deployment of persuasive interventions should be part of a general transport planning approach in cooperation with the transport authorities and various transport modes operators. An integrated approach where the transport system works for the benefit of travelers and persuasive technologies that support travelers' decisions could provide a significant impact [17].

Therefore, conscious mobility aims to improve environmental quality, foster intermodality, and enable data-driven decision making. This requires require new infrastructure policy and planning to accelerate the deployment of smarter-sustainable mobility. For the user to take advantage of these transportation alternatives, the reconfiguration of spatial distribution, the addition of emergent pathways, and the deployment of infrastructure for cleaner transport modes should be considered as part of a conscious infrastructure design. City planners and authorities would evaluate from a people centric perspective, the needs of the residents and then analyze the feasibility of infrastructure modification and integration to minimize environmental impacts. This design approach encourages conscious mobility behaviors while eliminating common constraints between users and sustainable modes of transportation.

The dimensions for conscious mobility are defined in Section 2 and establish the process of analysis for the case studies selected for this paper. Case studies selected were used to sample the analyze conscious mobility projects involving cleaner transport adoption, intermodality, and efficient transit systems as their main drivers.

*Conscious Mobility Dimensions*

From [1], the key smart and sustainability concepts were adopted for conceptualizing the dimensions of conscious mobility, focusing on user's mobility decision-making, technology incorporation and data collection, and commitment to public administration. As a result 5 dimensions of conscious mobility emerged:

1.  Society and Culture: Analysis of socioeconomic and cultural aspects of users that affect their decision-making process regarding their mobility patterns and their consequences.
2.  Infrastructure and Urban spaces: Construction, maintenance, and review of the physical space recognizing the current state of the road, the spatial distribution, and the new infrastructure to foster and deploy sustainable alternatives to transportation.
3.  Technology: Analysis and development of the elements needed for mobility operation, such as the modes of transportation (e.g., bus, car, bike, etc.) and the digital systems that gather and optimize releavnt data, increasing transportation resource efficiency and a better user experience.
4.  Normativity, Economy, and Politics: Support regarding policies, programs, or plans that dictate the paths towards sustainable transportation alternatives, foster mobility initiatives through their financial feasibility, and promote awareness about common externalities from economic activities.
5.  Environment: Analysis of the environmental and human-health negative consequences of the transportation sector, identifying possible mitigation solutions and measuring their impact.

## 3. Research Methodology

The methodology applied in this paper focuses on answering the following research question, "To what extent the city of Monterrey is prepared to embrace conscious mobility", the analysis is based in the transport modes connecting Distrito Tec, as well as behavior from users and the local community. Therefore, a systematic review of the literature was performed based on state-of-the-art (SOA) of sustainable, smart, electric, and conscious mobility concepts. The literature review presents a structured, explicit, and reproducible approach to the construction of a conscious mobility framework of analysis. Besides, an approach to the Soft Systems Methodology was used; this methodology attempts to foster learning and appreciation of the problem situation between a group of stakeholders rather than set out to solve a pre-defined problem. The complexity of many organizational/social problem context makes defining a problem a complex research task. Ultimately the challenge is how can we ensure people in our cities have access to and use more sustainable transportation options? [29]. It is important to clearly define the problem for each specific context of implementation to advance smart and clean innovation interventions in policy, technology, business models or social approaches to enhance sustainable urban mobility in Monterrey [30]. This paper proposes adding the need to assess the level of consciousness in urban mobility systems development and integration. Figure 2 presents the first conceptualization of conscious mobility, interconnecting the interrelationship and nexus between problems, improvements, and solutions analyzed.

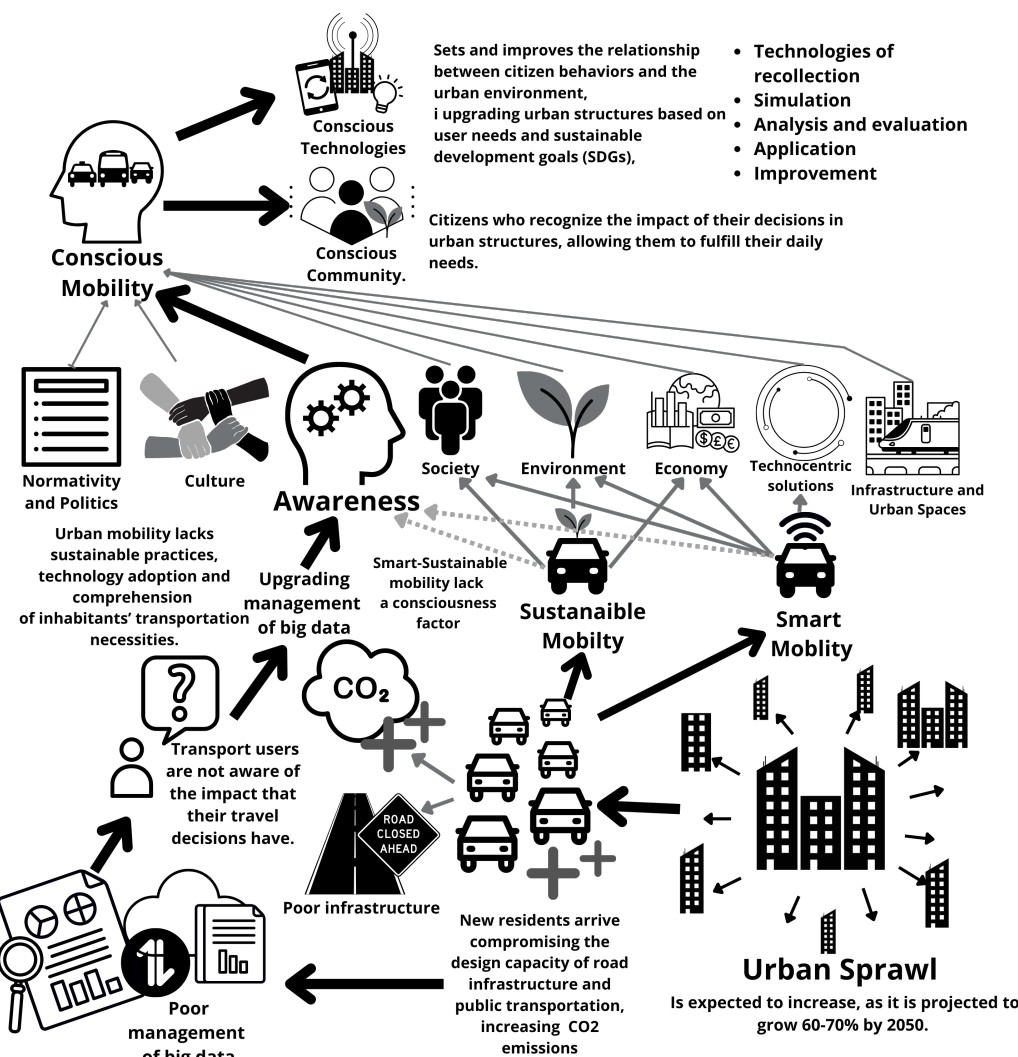

**Figure 2.** Soft methodology scheme of conscious mobility.

### 3.1. Research Criteria

A systematic literature review (SLR) identifies, selects, and critically appraises research to answer a clearly formulated question [31]. The SLR was performed through academic and scientific databases and publishers such as Google Scholar, MDPI, and Elsevier, among others. Data and technical reports from organizations, websites, and previously developed documentation were also retrieved, gathering quantitative and qualitative data for analytical and exemplification purposes.

The next parameters and baselines were considered to gather useful and relevant information for the research:

1.  Documents published from 2012 to 2022 were considered as part of the predetermined period of research, due to the high amount of public and private initiatives, with a focus on transportation systems, urban mobility, sustainable mobility, smart mobility, electric mobility, mobility behaviors, and the interaction of transportation systems with urban environments. Moreover, documentation published between 2017 and 2022 was prioritized regarding the SOA on transportation systems and technologies.
2.  Calzada, I. (2017) was referenced to review the state-of-the-art in mobility and conceptualize the conscious mobility dimensions [1].

3.  Documents [25,26] were selected as a repository for existing conscious mobility projects based on electric mobility due to their broad overview of cases in different stages and regions but specifically in America. However, individual case study data gathering was performed to fulfill missing information and contrast several points of view. Just the main case study was outside the repositories, selected to broaden the perspective of the objectives pursued by conscious mobility projects.

4.  Frameworks [32,33] were used to fill the gap within the selection and creation of suitable indicators for the case study of Monterrey. These frameworks were used just to extract and adapt indicators that best fit with the proposed dimensions of conscious mobility.

5.  Document [34] was used to understand the sustainable transport framework for the Metropolitan Monterrey Area (MMA), its challenges, and future strategies. Document [35] and its actualization [36] were also used to understand the current mobility status in DistritoTec and diagnose conscious mobility dimensions, laying the foundations for future conscious mobility projects.

6.  Given the qualitative approach of the research, extensive analysis of quantitative data was not a priority; besides, most of the databases consulted for Latin American countries are no longer available, or the information is not registered in a centralized database. It is intended that this research and final result serve as a skeleton and the starting point of discussion to initialize a correct data gathering process, which can be visualized by the community and ensure the correct feeding of the indicator framework.

### 3.2. Case Study Selection

This analysis considers five case studies to identify strategies and common paths for conscious mobility adoption. Cases were selected regarding five conditions:

1.  Allows for a comparison between conscious mobility projects based on electric mobility technologies, but in different contexts in America.

2.  Projects involved one or more outcomes of conscious mobility, such as expanding the accessibility of transport modes and making them more affordable and efficient, creating smart-sustainable transportation systems, and propitiating conscious infrastructure design in urban spaces.

3.  Inclusion of supportive technologies in the transport system of each city, related to the user's and stakeholder's decision-making. These were considered for the stage classification of the conscious mobility cases study.

4.  The projects are in different stages of development; in terms of adoption or implementation, thus the stage division proposed by the case study repository documentation [32] was used. However, it was applied in general terms regarding the five stages, so it can be applied to any initiative of conscious mobility.

    (a)   Stage 0: No substantial planning.
    (b)   Emerging stage 1: Talks and plans, but no pilot tests.
    (c)   Breakthrough stage 2: The city is running an initial pilot program.
    (d)   Growing stage 3: The city has gone past an initial pilot program.
    (e)   Consolidated stage 4: Mass adoption.

5.  The projects are well documented and reviewed within the repository documents selected.

The case studies selected are part of the actual efforts from the public and private sector to develop conscious mobility projects.

### 3.3. Case Study Analysis

Each case study was analyzed involving four steps to find coincidences or differences between the case studies to establish common paths toward the conclusions of the case as follows:

1. Recognizing mobility issues.
2. Reveling conscious mobility efforts.
3. Finding valuable indicators.
4. Concluding current mobility status.

The particular characteristics of the cases and their relationship to one another can be incorporated into the cross-case comparison component by a single group of researchers in a way that larger numbers of cases simply could not [37]. Therefore, thinking of mobility as a system, it was possible to observe and gather information about specific factors that can be compared among different cases. These factors were analyzed by applying the Environmental Dimensions of Transportation scheme [1], but expanding it to the five conscious mobility dimensions, identifying how inputs and outputs interact with each other, understanding what metrics should be assessed for each dimension of conscious mobility, and the current status of the city's transportation system (See Figure 3).

Stakeholder identification was still missing. Therefore, a framework for stakeholder engagement [38] was required to identify, analyze, and conclude which organizations, companies, and public entities were involved in the projects studied and what was their relationship to the fulfillment of the initiative. The conscious mobility matrix comparison is presented in Table A1 in Appendix A, giving a better understanding of the status of conscious mobility projects analyzed and the stakeholders involved.

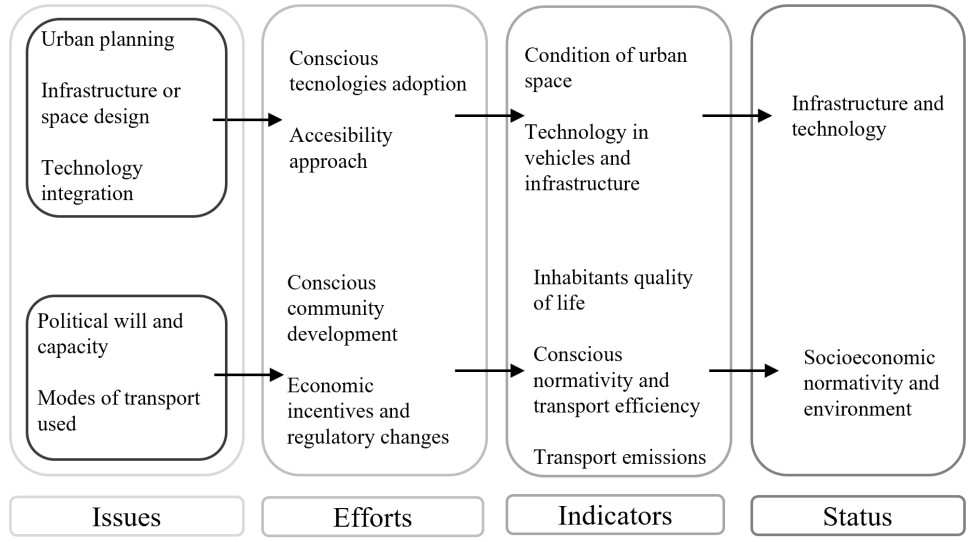

**Figure 3.** Analysis workflow of conscious mobility case studies.

*3.4. Conscious Mobility Indicators Framework*

The conscious mobility index framework was developed building on top of smart city frameworks [19,20] but focusing on human-centered mobility metrics as was meant to be achieved in the conscious mobility approach. Figure 4 presents the iterative process used for the design of the conscious mobility framework, first creating a general indicator framework about the metrics identified in the case studies selected for later adapting or creating useful indicators for the region of interest, in this case, DistritoTec.

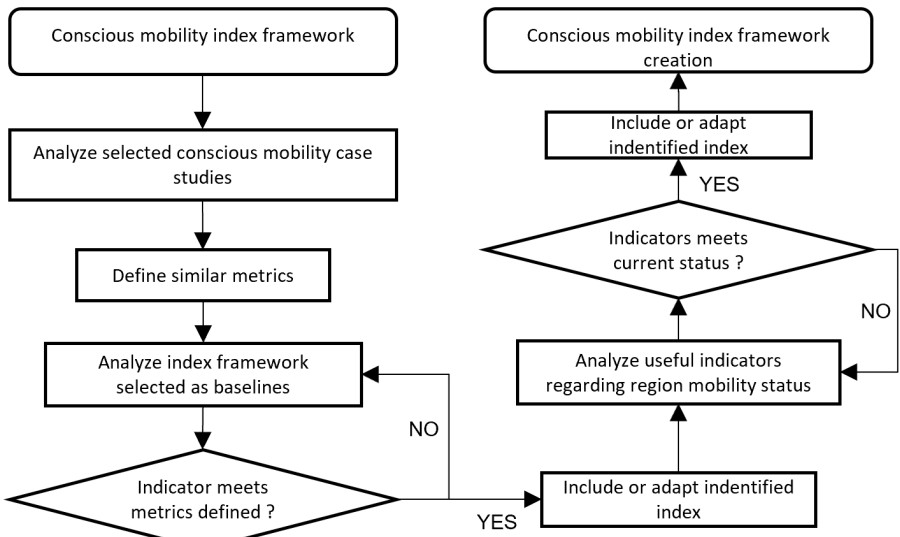

**Figure 4.** Flow diagram for the construction of conscious mobility indicators framework.

The initial application of this approach allowed for a better comprehension of the case studies analyzed and the metrics that should be included, as the iterative process allowed the contrast of the transport system status of each of the case studies with the indicators analyzed. Despite the efforts by international organizations to set common metrics, facilitating project status comparison and setting standardized indicators between the case studies was not possible at all, even though a preliminary index indicators and framework of analysis was created. Our hope is that researcher group replicate the methodological steps to build a conscious mobility framework for a particular urban context, therefore indicators are expected to differ depending on the background of the group and the goals of their research or project while maintaining the key parameters of the conscious mobility concept.

## 4. Case Studies

### 4.1. Bogota: BRT High-Quality Bus-Base Transit System

In the past two decades, Bogota had significant growth in private car ownership, causing high accident rates, long commuting times, and severe air pollution [39]. The city presents clear spatial segregation and unequal access to transport modes, the low-income groups live on the periphery, while the high-income people live close to the centers of activity [40]. The previous traditional public transportation system was dangerous, underused, and offered poor quality service. Moreover, it was operated with old bus fleets, required large amounts of fuel, and traveled at low average speeds; there were no designated bus stops, and trips were paid in cash on the bus [41].

The change was brought by TransMilenio, a Bus Rapid Transit (BRT) system part of the Integrated Public Transport System (ITPS) of Bogotá, Colombia [40]. BRT delivers fast, comfortable and cost-effective transport services at metro level capacities [42]. The project is a public–private partnership (PPP), in which responsibilities are distributed between the public sector (in charge of the investment in the infrastructure—stations, terminals, segregated lanes, etc.) and the private sector (responsible for the investments of the bus fleet, ticket selling, and validating the system and for the operation of the trunk and feeder services) [43]. Bogota's BRT is based on high-capacity buses operating in dedicated bus lanes on trunk routes, being supplied with passengers by feeder buses (which connect residential areas to BRT bus stops) [40]. Technology incorporation has also played an important role, as users' platforms and several management tools for collaborators enabled a better user experience, data analysis, and exchange of information, as well as opportunity areas identification [44].

Besides, the TransMilenio success, in 2012 Bogota began to explore cleaner and zero emission modes of transport through the Hybrid and Electric bus program, an initiative that aimed to ensure effective electrification and gradual replacement of all the internal combustion engine units [45]. Society has played an important role in supporting clean energy transportation technologies through social media campaigns, physical mediums, and online petitions [46]. As a result, the system has decreased the average travel time by 32%, increased property values along the main line by 15–20%, enhanced tax revenues, created jobs, and improved the health and safety of the community [39]. Further, as the allocation of public economic resources to support sustainable transport initiatives has been rising, EBs embracement has significantly increased, as 1061 units are already operating in the integrated transport system of Bogota, expecting to deploy 1485 by the end of 2022 [47].

Bogota´s conscious mobility project has been evolving to meet the increasing transport demand, ensure connectivity for all social sectors, and enhance the quality of life by taking action on health and environmental issues related to greenhouse gasses (GHG), emissions, noise pollution, and security. The accelerated embracement of electric mobility, Bogota is an example of good practices thanks to the development of synergies between the Government and different academic institutions to counteract aspects such as traffic congestion, at the same time the city has created a traffic management center through a monitoring platform examining the movements vials of the city and integrating captured data from cameras, traffic lights and bike paths; At this time the city, has started the tenders and changes in the intelligent signaling system and electronic tickets, trying to generate different alternatives to improve mobility and consolidate a safe and efficient transport for the city. This Center works hand in hand with the District Department of Mobility (SDM), which in coordination with the TransMilenio, the Transportation Terminal, and the Environment Secretariat, make decisions to manage and reduce response times in the incidents of mobility or eventualities that are presented, in addition an app called Moovit is granted that allows the population to plan the route they will make during the day, combining the TransMilenio with the integrated routes. Society is also concerned about health, accessibility, and environmental benefits, facilitating its acceptance. Therefore, Bogota is considered within stage 3 (growth stage) of conscious mobility classification, embracing conscious technologies for the optimization of mass transport but still missing complete electrification of the public transport system [48].

### 4.2. Columbus: Autonomous Vehicles for Equity

In 2016, the city of Columbus, Ohio, won the first U.S. Smart City Challenge created by the U.S. Department of Transportation (USDOT). Within their proposal, they recognized the challenges to disadvantage communities such as the neighborhoods of Linden that lack of transportation services presented. Linden and other similar neighborhoods had restricted access to quality jobs, health, and other basic services and also did not have the resources to use any private transportation service [49]. In addition, due to the high unemployment rate, the infant mortality rate was three times higher than anywhere else in the county.

As an improvement proposal, it was decided to create a network of connected electric autonomous vehicles (CEAVs) that would link these neighborhoods with the closest job centers, such as downtown Easton [49], which is one of the largest job centers in Ohio. This service would be provided through a smart card that was used to cover bus fees and car-sharing services, improving the economic integration of several families and enhancing the accessibility to health services and other essential services.

Through the implementation of the first autonomous shuttles (i.e., LEAP Shuttle), the Smart Columbus initiative was created, led by the city government and 2 mobility companies, May Mobility and EasyMile [50]. This initiative has the purpose of demonstrating how safe, clean mobility for all [49] can improve the life quality of its users. Within the initiative, a network of parallel projects was created [51], mostly focused on the Linden community, in which mobility strategies were proposed to address social problems using technology, which range from giving pregnant women access to health services and food

(Prenatal Trip Assistance), as well as the creation of various apps to improve daily life (Smart Columbus Operating System) and programs for the distribution of masks and meals during the pandemic through the CEAVs initiative.

An accessibility analysis [52] found that travelers originating at the Linden Transit Center can now reach at least 20,000 additional jobs and 3000 additional healthcare services within 30 min; this represents a clear improvement compared to travel time and accessibility to services before the introduction of the Smart Columbus projects. Beyond the tangible assets created by the grant, The Ohio State University calculated the investments from the implementation of the USDOT award generated an estimated gross metropolitan product (GMP) of USD 173.39 million and generated or induced 2366 jobs.

With the implementation of the Smart Columbus initiative, this demonstration project proved that salient equity issues faced by local vulnerable population communities can be addressed by providing state-of-the-art transportation systems to enhance access and mobility. Showing with performance metrics that novel integrated transportation options can empower residents to have a better quality of life. In this way, their ability to connect social problems with specific, measurable technological solutions is recognized [53]. Even though the project concluded its operation on April 2021, it is classified as a stage 2 (breakthrough stage) project of conscious mobility, where cutting-edge technologies are embraced by a community aware of their positive impacts and also part of the solution. The project further the pilot test enhancing food delivery during COVID-19 was not explored, but its aim demonstrates again the conscious mobility nature of its program goals [50].

*4.3. Shenzhen: Large-Scale Transport Electrification and Clean Economy Industrial Development*

Shenzhen emerged as a special economic zone, experiencing explosive growth by concentrating most of the foreign investment from the region, but this also brought drawbacks such as limited availability of land, rising labor costs, shortage of energy and water, and worsening environmental contamination [54]. In 2005 the government began to address the city's economic and environmental challenges, having identified the transportation sector as one of the main urban issues to tackle, and, in 2008, it was named a "City of Design" by UNESCO [54]. In this way, the government of China deployed a series of initiatives and incentives to decarbonize the transportation sector, beginning its path toward the electrification of public transport.

The complete electrification of the bus fleet in Shenzhen was consolidated in 2017, with a total of 16,539 EB, becoming the first city in the world to achieve it [55]. This success is recognized as an example of a joint effort between the private and public sectors, where the national government of China, together with the Shenzhen New Energy Vehicle Leading group (SNEVLG), were responsible for giving and managing economic incentives for the EB manufacturers, bus operating companies, and charging service providers, creating a stable ecosystem of electric mobility.

As part of this ecosystem, Shenzhen planned to upgrade the intelligent transportation system (ITS) in the following three stages: vehicle electrification and automation to vehicle connectivity and intelligentization, and ultimately shared-ride and integrated, intermodal transportation systems [56]. Stage 2 explores new mobility solutions to provide customized public transport services to the public and demonstrates the collaboration of electric mobility and smart mobility. It is planned to expand its mobile application further to integrate more urban mobility services to create a mobility-as-a-service (MaaS) platform [55]. Along with user experience improvements, a collaboration with the ride-hailing company Didi Chuxing has enabled access to a large amount of real-time traffic data, obtaining data such as passenger heat map to understand user usage of digital tools, route ridership, fare income, real-time vehicle movement, as well as real-time streaming of onboard cameras to make minor adjustments to the dispatch headway or resolve potential safety issues.

Thus, as technology was deployed systematically, several positive environmental, economic and social impacts have come from this transition. For instance data show, the preference of society for cleaner transport modes regarding environmental and national

energy security verified the effectiveness of the policy direction and environmental education from the central and local governments of China [57]. One indicator of the project's success is that the annual average pollutant concentrations in Shenzhen dropped from 2014 to 2019, in comparison with most Chinese cities, decreasing emissions of PM2.5, PM10, $NO_2$, and $SO_2$ [57]. Other positive impacts that are not deeply highlighted are related to the enhancement of accessibility, security, and odor conditions (related with the decrease of GHG emissions), perceived mostly in the urban core. A documented co-benefit of this major transport modes electrification endeavor has been making of Shenzhen the first quieter mega city [58].

Given the strong will from stakeholders of the private and public sectors to meet public health and environmental goals through the electrification of public transport, Shenzhen became the first case of a conscious mobility project successfully adopted in record time. Technology adoption has allowed public administration to offer better public transport services and gather useful data for system improvement. In addition, it is important to recognize the key role of inhabitants by fostering and accepting the widespread use of electric transport modes, and acknowledging the associated environmental benefits from this initiative. Moreover, a global industrial powerhouse in EVs and EBs has emerged along this process. Shenzhen-based BYD has become a global leader in high-tech electric mobility and innovation with the consequent positive impacts to the local economy. For instance, the creation of a local mobility innovation ecosystem and in the development of a clean economy supply chain and specialized talent clusters locally and around the world through subsidaries. Therefore, this case study can be considered in stage 4 (Consolidated), serving as an example of how sustainable-smarter modes can be fully integrated into the public transport system and have a major impact to local communities and industry.

### 4.4. Mexico City Bus Electrification and Active Mobility Programs for Sustainable Mega-Cities

Mexico City is one of the most populated urban centers in the world, with an estimated population of 22 million in its metropolitan area [59], representing huge mobility and transportation challenges. The increase in the vehicle fleet is estimated to be between 7 and 8 million vehicles, which has worsened the negative impact on the environment and society, representing 5% of the gross domestic product (GDP) of the city, including congestion, pollutant emission, noise, and accidents [60]. However, only 19% of the trips are made by private car, while 44% use public transport and 32% walk [44]. As can be seen, the solution to meet almost half of the mobility demand has been public transport, where motorized transport operates through 2 economic business models, the first one is highly subsidized public system, including the subway (i.e., Metro) and BRT systems, the second one based on leased concessions and made up of micro-buses (seat capacity of 7 to 30 riders) [60].

In 2005, it opened Metrobus, a BRT corridor along Avenue de Los Insurgentes [61], one of the longest street corridors in the world that crosses Mexico City, North-South. Metrobus serves roughly 12 miles of Insurgentes with 36 stations and 2 terminals. It replaced about 350 standard buses with 97 new articulated BRT vehicles. These vehicles dock at enclosed, rail-like stations, and passengers may enter or exit the vehicles at any one of four double-wide doors [45]. Fares are collected via automatic ticketing machines located at the entrance to stations; the fare is roughly USD 0.30, which enables passengers to travel any distance they choose along the corridor more cheaply [61].

Even though the BRT system was considered pretty efficient, in 2014, the Mobility law was enacted, aiming to achieve a more integrated, socially-inclusive, resistant, and people-focused mobility system, besides, in 2019, Mexico City's Strategic Mobility Plan was issue with the goal of redistributing the road space and promoting investments in sustainable methods of transport, improving infrastructures and services, and increasing accessibility and safety for all users [62].

That same year, the government announced its commitment to a 100% zero emissions corridor on Line 3 and Line 4 of the BTR system, which serves 168,000 and 65,000 passengers per day. As of 23 December 2020, a pilot project was carried out on Line 3 with the first

18-meter-long articulated bus, created specifically for Mexico City, by the EB manufacturer Yutong. Under the Metrobús Renewal Policy, it is expected that by 2030, all 81 units will be renewed. Further, in 2021, an electric bus pilot program was carried out on Line 4, implementing 12-meter-long Volvo 7900 model vehicles. These two pilot programs were monitored [59] by the International Council on Clean Transportation (ICCT) to collect operation data. Additionally, as part of the sustainable transport system, 290 electric trolleybuses are in operation [63], and the Ecobici system (bike-sharing mobility scheme) is the largest in Latin America, with 480 stations, 6000 bicycles, and 322 km of cycle paths [64].

The shift toward active mobility and electric mobility has meant a great advance in lowering pollutant concentration in the environment, where the BRT system avoids 122,000 of Co2 emissions every year [65], Ecobici avoided 4521 tons of Co2 between 2010 and 2019 [66], and a trolleybus fleet of 63 units avoided 79570 tons of Co2 every year [66]. Besides the positive environmental impact, important economic and health benefits have been achieved from the implementation of this sustainable transport system, such as USD 141 million in regained economic productivity as a result of travel time reductions from Metrobus Line 3 [67], the elimination of more than 2000 days of lost work due to illness, four new cases of chronic bronchitis, and two deaths per year saving the city an estimated USD 4.5 million [67] and an increase in passengers walking an average of 2.75 min more per day than before the city implemented its BRT system [67].

This case study represents a starting point for the prioritization of sustainable transport systems in an attempt to achieve zero-emission conscious mobility in Mexico City. The importance of this project lies in its ability to demonstrate that an integrated system of mobility can be a feasible alternative, having a great impact on mega cities, where negative externalities derived from transport activities are more salient to its citizens. On the other hand, it also highlights the correct application of policy regulations toward a sustainable-smart mobility. However, financial and technical capacity still represent a barrier to the massive adoption of the current pilot test. Moreover, the lack of supportive technologies to gather useful insights from big data mining for example that impede the correct identification of opportunity areas for improvement, thus, lowering the sustainable transportation efficiency potential of the system. Therefore, the Mexico City case study is classified as part of stage 3 of conscious mobility, hoping to reach stage 4 as financial and technical barriers can be overcome.

### 4.5. Monterrey: Quick, Safe & Sustainable Urban-Campus Community Mobility System

The metropolitan Monterrey area (MMA) is located in Nuevo Leon, Mexico, and is regarded as an industrial center with high development, representing 8% of the national Gross Domestic Product (GDP) in 2020 [68]. However, this has been accompanied by urban sprawl that has favored population growth, increasing 1.8 times from 1990 to 2019, experiencing a surface expansion of 2.7 times in the same period, lowering the density of the MMA [69]. As a result, mobility has become an important issue, given the excessive use of private cars, long commuting times, and lack of infrastructure to foster sustainable modes of transport [34], triggering negative impacts on society and the environment. In consequence, Tecnológico de Monterrey, a private university located in the south part of Monterrey, in collaboration with the Municipality of Monterrey, began the Partial Program for Urban Development (PPUD) in DistritoTec in 2013 (a district of urban regeneration around the university (See Figure 5).

The program aimed to diagnose the urban development status of DistritoTec, and propose several initiatives for their improvement. This case study was assessed by taking into consideration the five dimensions of conscious mobility, summarizing the whole picture of transportation issues on DistritoTec, and serving as a starting point for future projects of conscious mobility, specifically the incorporation of an EB in the local university mobility services "CircuitoTec", The system aims at providing a quick, safe and sustainable mode of transport for the Tecnologico de Monterrey comunity. Therefore, CircuitoTec plays an important role in the Tecnológico de Monterrey community having several strategic points

to help students, collaborators, and professors to transport quickly, safely, and sustainably around the campus [70].

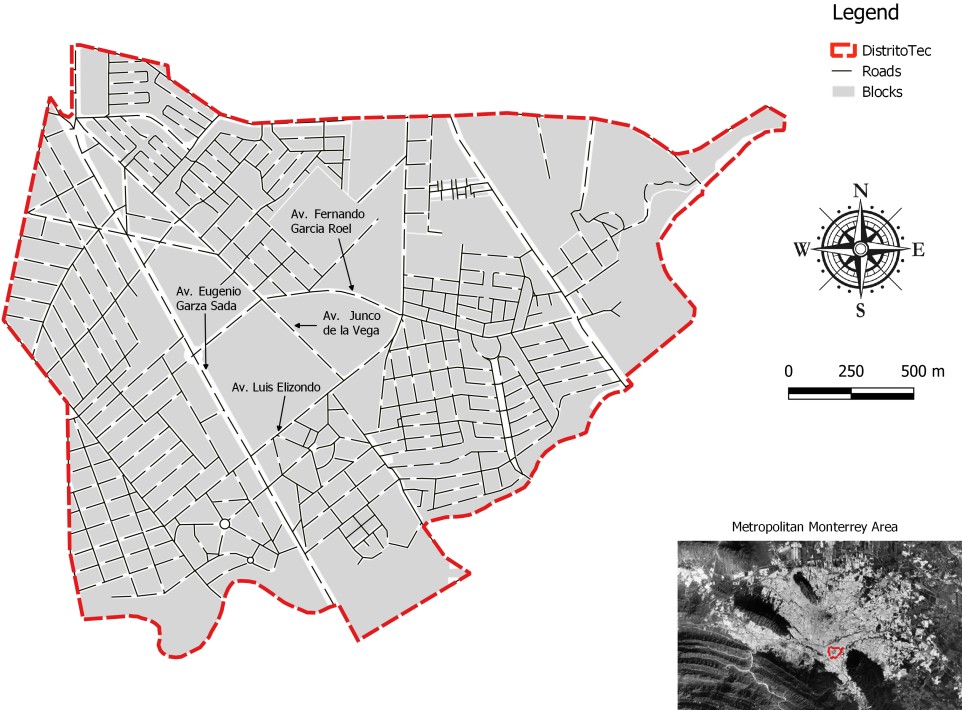

**Figure 5.** Localization of DistritoTec in the Metropolitan Monterrey Area.

### 4.5.1. Society and Culture

DistritoTec experienced a period of population decline between 2000 and 2010, going from 27,131 to 20,496 inhabitants [35]. Several factors led to this change, including urban sprawl, lack of housing opportunities for young families, cost of rent to student population, lack of services oriented towards students and public safety crisis from 2009 to 2013 [35]. These conditions led to a population preponderance of groups ranging 18 to 49 years old, followed by the senior citizens [35]. Given the predominance of students and an economically active population, local trips by active mobility modes such as walking represent almost a third of the modal share, but motorized transport still represents most of the modal mobility split, with a 48% share [35].

Most of these trips have as origin and destination the campus of the Tecnologico de Monterrey as the origin and destination, therefore a poll was conducted to understand the community´s mobility needs and behaviors to identify opportunity areas for university mobility services such as CircuitoTec. The survey poll provided useful insights, as general understanding from the community about what is CircuitoTec, however, more than 69% of the respondents answered that they did not use the service because of the comfort, time saved, flexibility, and facility that other modes of transport offer (e.g., shared-ride, private car, taxi, etc). There is a high awareness of what electric mobility is and its benefits, as more than 77% of the respondents know this mobility type, 99% would like it to be implemented in CircuitoTec, and also 85% said they would use CircuitoTec if it was clear to them that it was a sustainable mode of transport.

The purpose of this poll was to gather quick insights into the perception of the community of the transportation modes available, their travel decitions, and knowledge of sustainable modes of transportation. However, a follow up poll is planned to be conducted with a more significant sample and, focusing on a conscious mobility indicators details for its replicability in other urban community contexts.

### 4.5.2. Infrastructure and Urban Spaces

As part of the evaluation of infrastructure and urban spaces realized in 2014 [35], several aspects such as arborization, vegetation, streets, sidewalks, crossings, roundabouts, and storm drains were assessed regarding their accessibility, dimensions, signalization, state, presence, and deterioration. Considering the urban elements mentioned above, a classification system was developed ranging from A (good state) to D (very deficient state). Using this rating system, inner mobility paths from the DistritoTec were evaluated, with 11% of these in a good state (A), 19% regular or with deficiencies (B), 56% in a bad state (C), and 14% in a very deficient state (D).

The infrastructure quality analysis reflects limited accessibility by the poor conditions of principal corridors, getting even worse for inhabitants with limited mobility. Cycling presents the same problem because despite the current efforts to incorporate bicycle lanes to connect DistritoTec with the Center of Monterrey, there is not enough and well planned cycling infrastructure. Regarding public transport infrastructure, there are 35 current public transport bus routes in DistritoTec where there are 28 full bus stops (see Figure 6), 40 bus stops not signaled, and only 12 vertical signals [36]. Lack of the correct signalization, digitization and high-quality infrastructure represents common impediments to the adoption of smarter, more efficiente and sustainable transport modes.

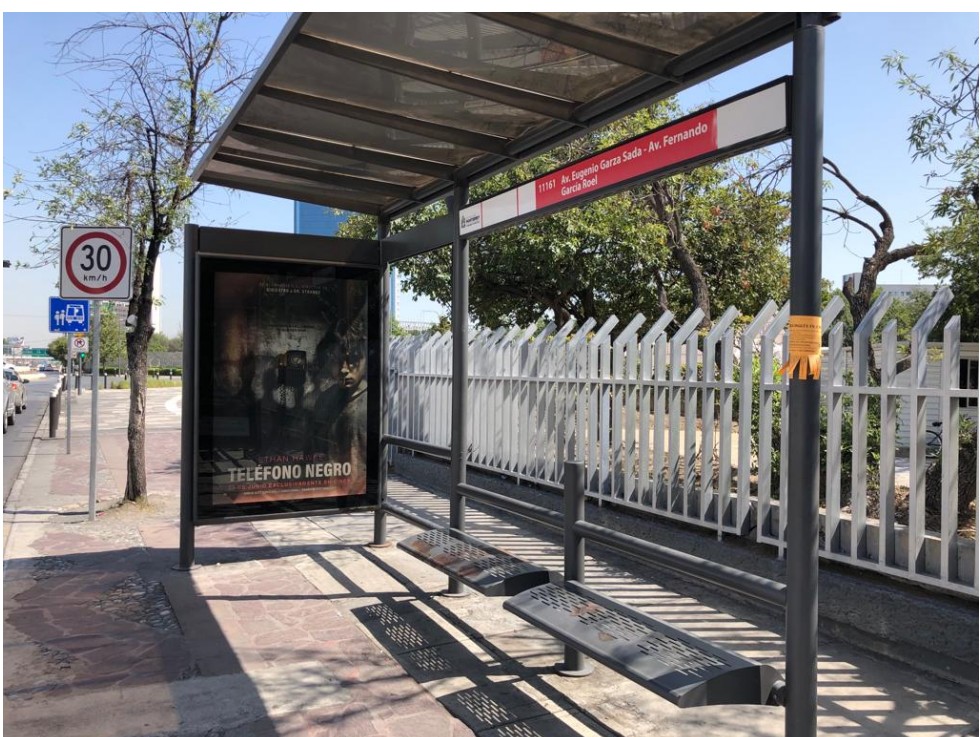

**Figure 6.** Full bus stop configuration in DistritoTec.

### 4.5.3. Technology

Clean and smarter technological evolution and adoption has been an important driver towards the improvement in public services that aim to increase the quality of life for all population sectors, particularly through environmental quality and sustainability outcomes. As part of the efforts within (MMA), the Metropolitan Integral Transit System (ISMT, spanish acronym) was one of the first technologies introduced in the city as a centralized system that operates in real-time to adjust traffic lights to manage congestion. However, its utilization has decreased given a lack of maintenance, updating and investment [34]. Also, buses fueled with compressed natural gas or electric charging stations have become an important stream of transportation technologies, since eight years ago as the incorporation of these advances contributes to mitigate GHG and co-pollutants [71].

DistritoTec has followed MMA´s path towards smarter-sustainable mobility encouraging emerging technologies, but focusing on micro-mobility. Preferential crossing for pedestrian on intersections around the campus and its urban surroundings has been enable through buttons connected to the traffic lights. Also, automated traffic lights time intervals adjustment occurs as pedestrian demand increases. A monitoring system has also been included in some intersections, as can be seen in Figure 7. Walking and cycling have also been encouraged by the incorporation of new LED street lights, increasing the range and quality of the brightness on streets. Finally, as an effort to facilitate electric mobility, EV charging infrastructure has been installed in several parking lots, and new projects to include electric units in the Tec de Monterrey mobility services. Also industrial alliance project for electrification of people transport systems with local industrial groups such as Questum mobility, Marcopolo, Daimler, the university (i.e,DistritoTec) and Senda, a private concessionary for staff and student transport services.

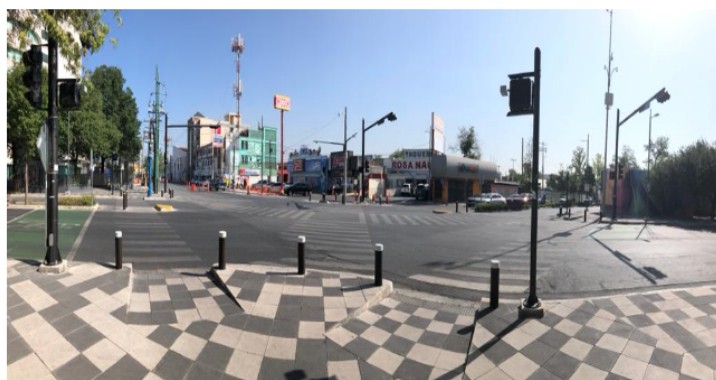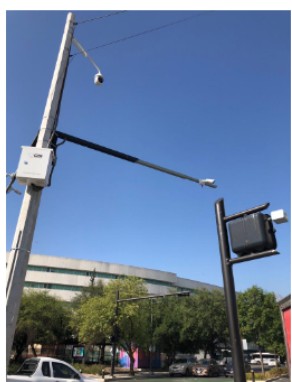

**Figure 7.** Complete streets and traffic monitoring system located within DistritoTec.

### 4.5.4. Normativity, Economy and Politics

As part of the current State administration (2021-2027), the Master's plan of Mobility contemplates a budget of almost 4900 million USD for the investment in infrastructure and mobility in four axes: dignified transport, smart mobility, complete streets, and regional connectivity [72]. Within these axes, several actions to advance sustainable and smarter mobility goals are described, focusing on the relocation of public transport routes, construction and maintenance of main roads, procurement of 1490 low emission and 110 electric units with an investment of 300 million USD, modernization of the ISMT and the center of mobility management, mobility sensor installation, and public spaces enhancement, with an investment of 41 million USD [72].

There is a strong political will toward sustainable mobility; proof of this is stated in the Sustainable Urban Mobility Program [34], a legal framework that aims to include the vision of urban mobility that the state of Nuevo Leon wishes to achieve in the next 10 years. The framework is integrated by the Sustainable Mobility and Accessibility law for the state of Nuevo León, the law of Climate Change of the State of Nuevo León, the law of Construction and Rehabilitation of Pavements of the state of Nuevo León, the law of Road Signs of the state of Nuevo León, and the state technical standard of sidewalks of Nuevo León.

At the municipality level, according to [73], several statements for the mobility initiatives of the DistritoTec are:

1. Maximize interconnection between routes to facilitate accessibility through walking.
2. Plan for equal access distribution of public space for different users.
3. Technologies adoption to enhance efficient management and reduce negative externalities.
4. Increase transport mode offers that fosters a reduction in private vehicle use.
5. Prevent accidents by, enhancing infrastructure and urban mobility.

6. Foster inclusion, in particular women's accessibility to mobility to minimize gender violence and enhance public safety.

7. Encourage new financial mechanisms and business models for the operation of public transport.

8. Include carpool systems, private-public transport, and fostering bicycle use.

In addition, DistritoTec, under the urban development plan of Monterrey´s municipality 2013–2025 (PDUMM, spanish acronym), is eligible to fulfill a series of actions for urban reordering, where urban mobility, intermodal transport, and public infrastructure play an important role. This plan led to holding several meetings between the public and private sectors, as well as the residents affected by the proposed projects designs. Even though inhabitants' perspectives were considered in the improvement of the projects, in some cases, these do not fulfill the necessities of the surrounding local community [74].

4.5.5. Environment

Monterrey experienced an increase in negative environmental externalities as a result of population growth, a rise provate, public and logistics combustion vehicles use, and growth in local industrial and commercial activity [35]. In the MMA, private vehicles and taxis are major contributors to atmospheric pollution, generating 70% of the emissions of carbon monoxide (CO), volatile organic compounds (COV), and nitrogen oxides (NOx), but have a minor inference on particulate matter (PM) [75]. Therefore, measuring the emissions for each type of pollutant has become an important issue to explore, leveraging technological advances for the automatization of this process. To address this issue Monterrey City began the deployment of its environmental monitoring integral system (ISEM, spanish acronym). It consists of 14 emission monitoring stations located around the MMA and represents an important tool for monitoring and reporting pollution concentration of CO, PM10, and PM2.5 emissions (see Figure 8) [76]. As part of the annual air quality monitoring program, for instance in 2019 before the COVID19 public health crisis, 181 days were reported to fail to meet local and federal air quality standards based on federal health and air pollution regulatory parameters [36]. Therefore, there is a clear nead to characterize the relationship between air pollution emissions generated and each transportation mode, enabling correcting measures to address local pollution and planetary health. This would constitute a step forward towards conscious mobility in Monterrey.

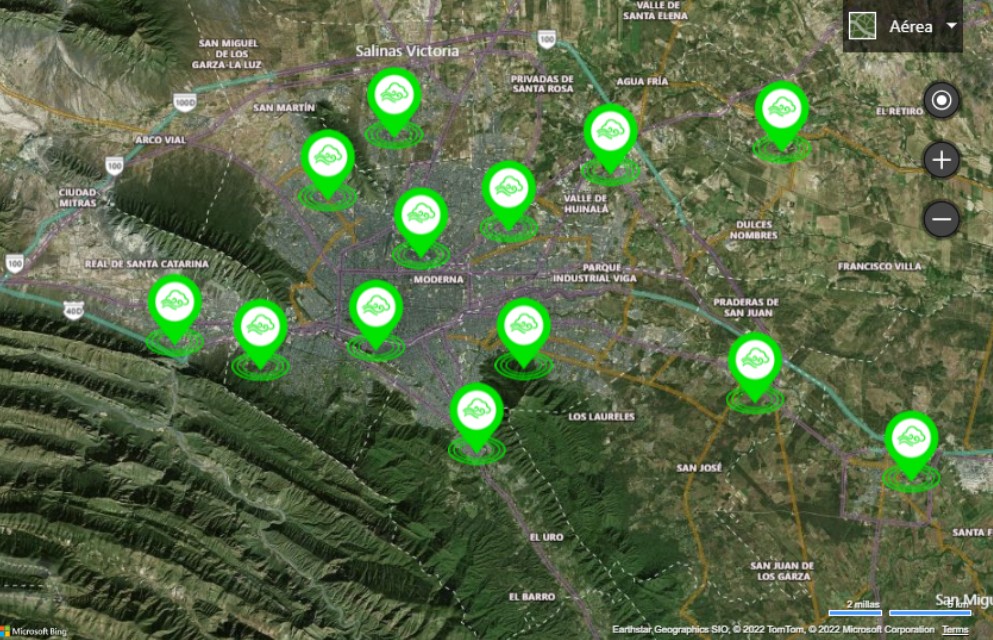

**Figure 8.** Active station part of the integral system for environmental monitoring in MMA.

### 4.5.6. Conclusion of the Case Study

After the analysis of the five dimensions of conscious mobility, DistritoTec seems to be going through a period of innovation and technological embracement, rethinking the way a transportation system should work, which modes of transportation need to be fostered, and what is necessary to improve citizen's quality of life while minimizing negative transportation externalities. Strong political will and a good allocation of investment are playing an important role in the development of sustainable-smart mobility strategies, leading toward the first approach to conscious mobility. Therefore, as the adoption of technology to improve stakeholders decision-making is still in its nascent steps, and the community is just starting to be more aware of the benefits of sustainable modes of transport and a conscious design of infrastructure, this case study is classified within stage 1 (emerging stage).

## 5. Results and Discussion

We hope the lens that the concept of conscious mobility enhances future awareness of the complex web of factors and interconnections and nexus between social, environmental and physical and systems in the process of adopting smarter and more sustainable mobility systems to minimize the negative externalities from conventional transportation modes

The multi-case study analysis performed in Section 4 allowed to identify the key aspects involved in conscious mobility projects regarding the four steps followed in conducting the analysis. Given the increased societal concern on the environmental quality and its relationship with health issues and climate risk, this is cataloged as the main problem to address by the current conscious mobility projects, but, technology integration, lack of sustainable normativity frameworks and poor accessibility are also important issues considered. Therefore, even though the development of compressed natural gas projects (CNG) have bring substantial benefits such as quality of air improvement and mitigation of co-pollutans, the electrification of mobility projects has been the preferred sustainable alternative to incorporate, as the future incorporation of zero emission vehicles (ZEVs) could lower drastically, emissions in the transport sector while bringing great benefits to citizens and the environment. A key factor will be that electric power sources that energize the (EV) charging infrastructure become clean to close such virtual loop.

The accelerated transition toward electric mobility has fostered several related efforts such as policy drivers, economic incentives, more conscious infrastructure design, public–private partnership, and the integration of enabling technologies to enhance operational efficiency and a better user experience for society. However, this transition has not advanced at the same rate in all regions as the case studies reflect. Thus, the importance of the conscious mobility indicators framework as a customized and flexible tool to follow best and smart practice in successful or leading conscious mobility projects. The framework allows to, identify those indicators that best align with the current mobility status of the city to be assessed. Table 2 presents the conscious mobility indicators framework for Monterrey's DistritoTec, setting a first step towards the design of a framework for the development of conscious mobility projects in the MMA region and elsewhere.

**Table 2.** Conscious mobility index framework for DistritoTec, Monterrey

| Dimension | Indicator | Metric | Description |
|---|---|---|---|
| Society and Culture | Traffic accidents | #/100,000 | Number of transportation fatalities per 100,000 population |
| | Accessibility to public transport | likert | Perception from the users about the accessibility to public transportations modes |
| | Population with College education | % of people | Percentage of people with college education |
| | Use of public transport modes and active mobility | % of mode share | Percentage of total trips made by public transportation alternatives or active mobility |
| Infrastructure and Urban Spaces | Public transport infrastructure | % in km | Length of road surface where public transport can transit |
| | Length of bike route network | % in km | Proportion of bicycle paths and lanes in relation to the length of streets (excluding motorways) |
| | Length of sidewalks | % in km | % of sidewalks in relation to the length of streets |
| | Configuration of the space | likert | Absence of exclusive lanes, amenities, street lights, and anything that foster active mobility |
| | Road surface conditions | likert | Condition of the road surface regarding potholes and cracks |
| Technology | Availability of traffic monitoring using ICT | % of streets | Length of streets with traffic monitoring |
| | Intersection control | % of intersections | Number of crossroads with adaptive traffic light duration |
| | Low-carbon emission passenger vehicles | % of vehicles | Percentage of low-carbon emission passenger vehicles |
| Normativity, Economy, and Politics | Availability of transportation modes (for region, in a city) | from 0 to # | Transport modes available in Monterrey, in a scale from 1 to #, being # the number of public transport modes. |
| | Share of mobility public procurement | % in M USD | Percentage annual procurement of mobility investment as share of total annual procurement of the city administration |
| | Incomes from public passenger vehicles trips | % of revenue | revenue with respect to the total cost of the operation in the present year |
| | Travel time | % in hours | Increase in overall travel times when compared to free flow situation uncongested situation |
| | Peak hours lapses | Ratio/value | Ratio of travel time during the peak periods to travel time at free flow periods |
| | Citizen participation | % of projects | Number of projects in which citizens actively participated as a percentage of the total projects executed |
| | Conscious mobility | Likert | Extent to which the city has a supportive conscious mobility policy |
| Environment | Annual final energy consumption | MWh/cap/yr | Annual final energy consumption for transportation sector |
| | Air quality index | Danger level associated | Annual concentration of relevant air pollutants, ranging from good to extremely bad |
| | Amount of air monitoring stations | # stations | Number of stations regarding the suggested quantity |

The above indicators are expected to evolve and include several more indicators as technology advances, political will, and societal awareness increase. Community en-

gagement in project planning, design and implementation seems to be a critical factor to this process. In Part, project consciousness is perceived to be strongly related to the level of active involvement and engagement of stakeholders, rather than the region where the project was implemented. For example, high level participation from national private sector and national government, it was supportive of conscious mobility aspects and successful performance of the project, as in the case of Columbus Ohio (i.e., LIFT shuttle) and Shenzhen (i.e., EV/EB deployment and world class innovation and production ecosystem). However, local government and local communities engagement was critical across all case studies. Table 3 presents stakeholders involved in each project, divided into seven groups.

**Table 3.** Stakeholders involved in conscious mobility projects by case study.

| City | Local Government | National Government | Mobility Companies | EB Manufacturers | International Organization | Academia | Community or Associations |
|---|---|---|---|---|---|---|---|
| Bogotá | ✓ | | | ✓ | ✓ | | ✓ |
| Colombus | ✓ | ✓ | ✓ | | | ✓ | ✓ |
| Shenzhen | ✓ | ✓ | ✓ | ✓ | | | ✓ |
| Mexico City | ✓ | | | ✓ | ✓ | | ✓ |
| Monterrey | ✓ | | ✓ | | | ✓ | ✓ |

After the mapping of stakeholders, it is observed a balanced diversity of conscious mobility actors, helping better understanding their involvement and influence on smarter, more conscious and sustainable mobility project success. In the case of Mexico City and Bogota, even though they have actors involved and similar legal frameworks oriented toward sustainable public transport, the level of adoption is quite different since Mexico City has barely begun its transition toward electric mobility. After successfully completing the pilot test in 2022, 9 more EBs units were introduced into the BRT system, equivalent to 6.6% of the total fleet [77,78] and 500 trolleybuses are expected to be in operation by the end of 2024, while the city of Bogota achieved its first milestone towards full electrification of the public transport by acquiring 1485 EBs. Conscious enabling technologies (e.g., digitization, data mining, artificial intelligence and optimization, etc.) have presented a clear advance among all the case studies, from convenient fare payment systems to route planning and other mobility visualization experiences of the users and improving transport system operation and efficiency.

## 6. Conclusions

The concept of conscious mobility emerged as a way to better integrate conventional sustainable and smart mobility planning considerations with behavioral decision making and choice by services communities. It aims to leverage user travel decisions, values and concerns (e.g, health, convenience, sustainability, equity, safety, cost-effective, etc) to better inform government and business decisions in the provision of smarter, more sustainable mobility services.Therefore, the purpose of this research was to present an exploration of key user awareness indicators towards a robust conceptualization of "conscious mobility", both as a performance parameter to urban transport technologies and infrastructure deployment and as a more holistic framework of analysis. This novel perspective is still in the early stages of development. The case studies analyzed here, show that valued and concerns individual awareness in travel decisions in urban structures and technologies that seek to improve mobility services are increasing, while reducing environmental and health issues are being deployed at different scales and scopes, from fully-electrified mega city public transport systems to a targeted community pilot to enhance equity, sustainability and access through cutting edge technologies. In responding our research question Monterrey´s Distrito Tec´s urban campus community is prepared to lead in using conscious mobility analysis to enhance transport systems planning and design for its community and serve as a demonstration pilot for the rest of MMA. There seems to be a strong commitment and

momentum among key factors, from local authorities, private sector, academy, to the local community in general, to overcome these type of multidimensional mobility projects.

**Author Contributions:** Conceptualization, R.C.V.-M., J.G.L.-R., J.N.-B., A.V.-M. and J.d.J.L.-S.; Investigation, R.C.V.-M. and A.V.-M.; Methodology, R.C.V.-M.; Project administration, R.C.V.-M.; Supervision, M.A.R.-M., J.N.-B., R.A.R.-M., B.L.P.-H., A.V.-M., J.F.J.V and J.d.J.L.-S.; Validation, J.N.-B.; Writing—review & editing, R.C.V.-M. and J.G.L.-R. All authors have read and agreed to the published version of the manuscript.

**Funding:** This research received funding by the CampusCity Initiative of the Tecnologico de Monterrey.

**Informed Consent Statement:** Informed consent was obtained from all subjects involved in the study.

**Data Availability Statement:** Not applicable.

**Conflicts of Interest:** The authors declare no conflict of interest.

## Appendix A

**Table A1.** Conscious mobility case studies matrix comparison.

| City | Country | Conscious Technologies | Status | Issues | Efforts | Stakeholders Involved | Project Objective |
|---|---|---|---|---|---|---|---|
| Bogotá | Colombia | Recollection, Analysis and Evaluation and Application | Stage 3 operating | Transport system efficiency Poor accessibility Financial and technical capacity Pollution Lack of normativity | Technology acquisition Monitor Air quality Establish private-public partnerships Economic incentives and policy enacted | Local government EB manufacturers International organizations citizens and associations | Air quality improvement and increase transportation accessibility |
| Colombus | United States of America | Recollection and Application | Stage 2 project ended | Poor accessibility Lack of normativity | Technology acquisition Establish private-public partnerships Economic incentives | Local government National government Mobility companies Academia Citizens and associations | Enhance accessibility to basic services, increase job opportunity and reduce child mortality |
| Shenzhen | China | Recollection, Simulation, Analysis and Evaluation and Application | Stage 4 operating | Transport system efficiency Pollution | Technology acquisition Monitor Air quality Establish private-public partnerships Economic incentives and policy enacted | Local government National government Mobility companies EB manufacturers Citizens and associations | Reducing oil dependency, strengthening automotive sector and improving air quality |
| Mexico City | México | Analysis and Evaluation, and Application | Stage 3 ending pilot | Transport system efficiency Pollution Financial and technical capacity Lack of normativity | Technology acquisition Conscious infrastructure design Monitor Air quality Establish private-public partnerships Policy enacted | Local government EB manufacturers International organizations Citizens and associations | Create a zero emission zone and green corridor. |
| Monterrey | México | Application | Stage 1 not in operation | Transport system efficiency Poor accessibility Lack of normativity | Conscious infrastructure design Monitor Air quality Establish private-public partnerships Policy enacted | Local government Mobility companies Academia Citizens and associations | Enhance transportation quality services and air quality |

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
