# Peer review of "Conscious Mobility for Urban Spaces: Case Studies Review and Indicator Framework Design"

_applsci, doi:10.3390/app13010333_

Round 1
Reviewer 1 Report
Comments to the Authors:
Overview:
The present study aimed to define the concept of conscious mobility and its dimensions. The study tried to assess the extent to which the the city of Monterrey is prepared to adopt conscious mobility. The topic analyzed in the paper is relevant and important, however, the methodology could be described in more detail.
The novelty and advantages of this study should be highlighted in comparison with previous ones.
Major comments:
1. The main concern I have about the paper is related to the Methodology. In the methods section, the stages of the study could be described and explained by providing a scheme or diagram.
2. The selection of the respondents for a pool should be described in more detail. How was the pool conducted? How the respondents were contacted and involved in the pool? What was the response rate? How the sampling was conducted?
3. Also, what was the inclusion and exclusion criteria for the respondents? The characteristics and behavior/lifestyle factors of the respondents should be described in detail and presented in the manuscript.
4. There is a lack of information on socioeconomic and demographic variables and their influence on mobility behavior.
5. The quality of figure 1 and 2 should be improved.
Author Response
Responses to Reviewer 1
Thank you for your valuable feedback, we will present the responses to your suggestions:
1. The main concern I have about the paper is related to the Methodology. In the methods section, the stages of the study could be described and explained by providing a scheme or diagram.
Response 1
A diagram to summarize and have a better perspective of the steps followed in the methodology will be included
2. The selection of the respondents for a pool should be described in more detail. How was the pool conducted? How the respondents were contacted and involved in the pool? What was the response rate? How the sampling was conducted?
Response 2
We will provide more details about this first poll, however a second poll with more respondants and a deeper research is intended to be conducted in a second paper where the study will be focus in Monterrey
3. Also, what was the inclusion and exclusion criteria for the respondents? The characteristics and behavior/lifestyle factors of the respondents should be described in detail and presented in the manuscript.
Response 3
The groups of respondants are going to be described with a little more detail, but it is not the objetive of this study to segment in great detail the respondants of the poll, it was just considered to include the main three groups that are part of the Tecnologico de Monterrey community
4. There is a lack of information on socioeconomic and demographic variables and their influence on mobility behavior.
Response 4
Sociodemographic behavior will be included in the case studies and also this relationship will be described in the other sections of the paper.
5. The quality of figure 1 and 2 should be improved.
Response 5
The quality was already improved.
Reviewer 2 Report
This research presents a Conscious Mobility study, where the authors presents the definition of conscious mobility, its main components, dimensions and also, they present an approach to evaluate this concept in current mobility projects. In this sense, the paper presents an interesting research work, however, the paper contains several issues that need to be solved to increase the paper quality.
The introduction section provides enough background the project, however, this section didn’t present the main objective of the research work, which is needed to clarify the rest of section of the paper. The paper didn’t define the contributions of the presented research work compare with the state of the art.
It is recommended for the authors to include the research objectives in the introduction section, as well as to clearly define the specific issues that are solved by the proposed solution. The conclusion section mentioned the objective but only in the surface, it is needed to include this in the introduction section to enable the reads to understand, from the beginning, the research goal that are intended to achieve in the research work.
It is very important to include in the paper a state of the art that stablish alternatives solutions proposed by other research groups. That state of the art will permit to give value to the contributions of the study presented in the paper.
The title of the paper needs to be revised because the paper is not oriented to shown a Framework, but to presents a study of the mobility in a city. Also, it is important to verify the appropriated match among title, the research questions, results and objective. Currently, inconsistencies can be found among the paper concepts.
In the presented case studies, no details are given to show the manner in which the mobility dimensions are evaluated. The authors need to detail the dimensions that were evaluated in each case study. Also, the author didn`t justify why the presented case studies represent a conscious mobility example, no quantitative evidence is presented to justify the evaluation approach.
The paper doesn’t describe the digital technologies to collect information in the analyzed case studies.
The paper mentioned that a systematic review was performed but no information is given about this review. In order to appropriately present the systematic review, the authors need to include the research question, the data sources analyzed, the inclusion/exclusion criteria and the results of the review.
A carefully review of the paper references need to be completed by the authors because some of the references are incomplete o out of date.
Author Response
Responses to Reviewer 2
Thank you for your valuable feedback, we will present the responses to your suggestions:
1. The introduction section provides enough background the project, however, this section didn’t present the main objective of the research work, which is needed to clarify the rest of section of the paper. The paper didn’t define the contributions of the presented research work compare with the state of the art.
Response 1
The main objetive and second objetives will be described in a more detailed way. Also the state of the art of mobilty as smart mobility, autonomous mobility and the little research of conscious mobility will be included an stated how this paper present a different perspective that should be analyzed.
2. It is recommended for the authors to include the research objectives in the introduction section, as well as to clearly define the specific issues that are solved by the proposed solution. The conclusion section mentioned the objective but only in the surface, it is needed to include this in the introduction section to enable the reads to understand, from the beginning, the research goal that are intended to achieve in the research work.
Response 2
The solution/contribution that present this study will be described in a more detail way.
3. It is very important to include in the paper a state of the art that stablish alternatives solutions proposed by other research groups. That state of the art will permit to give value to the contributions of the study presented in the paper.
Response 3
State of the art will be included.
4. The title of the paper needs to be revised because the paper is not oriented to shown a Framework, but to presents a study of the mobility in a city. Also, it is important to verify the appropriated match among title, the research questions, results and objective. Currently, inconsistencies can be found among the paper concepts.
Response 4
The tittle of the paper will be analyzed to have a better integration between the researc question, results and objetives.
5. In the presented case studies, no details are given to show the manner in which the mobility dimensions are evaluated. The authors need to detail the dimensions that were evaluated in each case study. Also, the author didn`t justify why the presented case studies represent a conscious mobility example, no quantitative evidence is presented to justify the evaluation approach.
Response 5
The aim of this study was to present a first conceptualization of a how a conscious mobility in urban spaces could be assessed, however it was meant to analyze the case studies in a more qualitative way, until the conscious mobility dimensions and the indicators can be weighed. The dimensions perceived to be present in the case studies and why they represent a conscious mobility example will be explained in a more detailed way.
6. The paper doesn’t describe the digital technologies to collect information in the analyzed case studies.
Response 6
In most of the cases there are not much information about how the data was collected, and some of the public databases where this data should be available are no longer in function. This is a common problem within the Latin America region and we could describe it in a more detail way to understand why this information is missing.
7. The paper mentioned that a systematic review was performed but no information is given about this review. In order to appropriately present the systematic review, the authors need to include the research question, the data sources analyzed, the inclusion/exclusion criteria and the results of the review.
Response 7
The selected criteria in the study will be reviewed to be specified in the research criteria subsection.
8. A carefully review of the paper references need to be completed by the authors because some of the references are incomplete o out of date.
Response 8
References will be reviewed.
Reviewer 3 Report
The manuscript presents a newly developed concept namely "Conscious Mobility" for the city of Monterrey based on mobility-related matrices of five different cases. The manuscript is well-written and concerns an important topic. However, my main concern is the assessment approach: the five case studies are summarized qualitatively, and the analysis flow and mobility indicators' framework look ambiguous to me. I suggest the authors follow some standard qualitative assessment methods to come up with the assessment matrices. For instance, the authors could consider the "Soft System Methodology", which aims to find a better way of dealing with a situation about which we think that "something needs to be improved about that". In this regard, much can be found regarding the "Stafford Beer's Viable System Model" that the authors could make use of to improve their approach.
Other minor remarks:
It's difficult to follow some of the abbreviations in the paper. For instance, Line 93: The abbreviation "ICT" comes for the first time and hence needs a full form; Line 464: What is ZMM? I suggest the authors write the full form of abbreviations that only appear two or three times in the paper.
The objectives are not clearly mentioned. The authors need to specify the main objectives clearly in the introduction part.
Author Response
Responses to Reviewer 3
Thank you for your valuable feedback, we will present the responses to your suggestions:
1. The manuscript presents a newly developed concept namely "Conscious Mobility" for the city of Monterrey based on mobility-related matrices of five different cases. The manuscript is well-written and concerns an important topic. However, my main concern is the assessment approach: the five case studies are summarized qualitatively, and the analysis flow and mobility indicators' framework look ambiguous to me. I suggest the authors follow some standard qualitative assessment methods to come up with the assessment matrices. For instance, the authors could consider the "Soft System Methodology", which aims to find a better way of dealing with a situation about which we think that "something needs to be improved about that". In this regard, much can be found regarding the "Stafford Beer's Viable System Model" that the authors could make use of to improve their approach.
Response 1
Methodology will be evaluated to present a diagram where it can be easier to understand how the analysis was conducted. This first approach about how conscious mobility can be assessed is more qualitative despite the indicators for each dimension are included, this is because weighing the indicators will involve a deeper analysis that is meant to be included in a second study of conscious mobility. We will consider the suggested methodologies to improve our analysis.
2. It's difficult to follow some of the abbreviations in the paper. For instance, Line 93: The abbreviation "ICT" comes for the first time and hence needs a full form; Line 464: What is ZMM? I suggest the authors write the full form of abbreviations that only appear two or three times in the paper.
Response 2
Abbreviations were already corrected.
3. The objectives are not clearly mentioned. The authors need to specify the main objectives clearly in the introduction part.
Response 3
Objectives will be clarified.
Round 2
Reviewer 3 Report
I cannot see the modifications made by the authors in the revised version. The authors should highlight the changes they have made in the manuscript following my comments.
After comparing the old and new versions, I can see that they have reflected my comments #2 and #3. However, I still do not see, how Comment #1 was reflected in the revisions. I am copying pasting the comment here again:
"However, my main concern is the assessment approach: the five case studies are summarized qualitatively, and the analysis flow and mobility indicators' framework look ambiguous to me. I suggest the authors follow some standard qualitative assessment methods to come up with the assessment matrices. For instance, the authors could consider the "Soft System Methodology", which aims to find a better way of dealing with a situation about which we think that "something needs to be improved about that". In this regard, much can be found regarding the "Stafford Beer's Viable System Model" that the authors could make use of to improve their approach."
Author Response
The comment #1 is addressed in the document attached. Below the corrections made can be seen.
|
Comment |
Color |
Section |
Page |
|
Number 1 Methodology |
Turquoise |
Introduction, Research Methodology |
2, 5, 6, 7 and 8 |
|
Number 2 abbreviations |
Yellow |
Conscious mobility, Research Methodology, Case Studies |
3, 6, 12, 14 and 16 |
|
Number 3 Objectives |
Green |
Introduction |
2 and 3 |
